# Preparation of Soluble Complex of Curcumin for the Potential Antagonistic Effects on Human Colorectal Adenocarcinoma Cells

**DOI:** 10.3390/ph14090939

**Published:** 2021-09-19

**Authors:** Jamal Moideen Muthu Mohamed, Ali Alqahtani, Barkat A. Khan, Adel Al Fatease, Taha Alqahtani, Krishnaraju Venkatesan, Fazil Ahmad, Bashar I. Alzghoul, Ali Alamri

**Affiliations:** 1College of Pharmacy, Shri Indra Ganesan Institute of Medical Science, Tiruchirappalli 620012, Tamil Nadu, India; 2Department of Pharmacology, College of Pharmacy, King Khalid University, Guraiger, Abha 62529, Saudi Arabia; amsfr@kku.edu.sa (A.A.); Ttaha@kku.edu.sa (T.A.); krishcology@gmail.com (K.V.); 3Drug Delivery and Cosmetics Lab (DDCL), GCPS, Faculty of Pharmacy, Gomal University, Dera Ismail Khan 29050, Pakistan; barkat.khan@gu.edu.pk; 4Department of Pharmaceutics, College of Pharmacy, King Khalid University, Guraiger, Abha 62529, Saudi Arabia; afatease@kku.edu.sa (A.A.F.); aamri@kku.edu.sa (A.A.); 5Department of Anesthesia Technology, College of Applied Medical Sciences in Jubail, Imam Abdulrahman Bin Faisal University, P.O. Box 4030, Jubail 35816, Saudi Arabia; fmahmad@iau.edu.sa; 6Respiratory Care Department, College of Applied Medical Sciences in Jubail, Imam Abdulrahman Bin Faisal University, Dammam, Jubail 35816, Saudi Arabia; bialzghoul@iau.edu.sa

**Keywords:** curcumin, colorectal cancer, hot-melt extrusion, phase solubility study, in silico molecular study, central composite design

## Abstract

This study was designed to investigate the effects of curcumin (CMN) soluble complex (SC) prepared by melt casting (HM) and hot-melt extrusion (HME) technology. Phase solubility (PS) study, in silico molecular modeling, aqueous solubility, drug release, and physicochemical investigation including a novel dyeing test was performed to obtain an optimized complex by a central composite design (CCD). The results show that the HME-SC produces better improvements towards solubility (0.852 ± 0.02), dissolution (91.87 ± 0.21% at 30 min), with an ideal stability constant (309 and 377 M^−1^ at 25 and 37 °C, respectively) and exhibits A_L_ type of isotherm indicating 1:1 stoichiometry. Intermolecular hydrogen bonding involves the formation of SC, which does not undergo any chemical modification, followed by the complete conversion of the amorphous form which was identified by XRD. The in vitro cytotoxicity showed that IC_50_ was achieved in the SW480 (72 µM.mL^−1^) and Caco-2 (40 µM.mL^−1^) cells while that of pure CMN ranged from 146 to 116 µM/mL^−1^. Apoptosis studies showed that cell death is primarily due to apoptosis, with a low rate of necrosis. In vivo toxicity, confirmed by the zebrafish model, exhibited the safety of the HME-SC. In conclusion, the HME-SC potentially enhances the solubility and cytotoxicity to the treatment of colorectal cancer (CRC).

## 1. Introduction

A tumor existing in the colon or rectum is called colon cancer or rectal cancer, depending upon the affected parts of the GIT. CRC is the third highest cause of common cancer in males, the second-highest cause of common cancer in women, and the death rate is third among all types of cancers worldwide [1]. In 2020, nearly 147,950 people were diagnosed with CRC and 53,200 died from the disease. This includes 17,930 cases and 3640 deaths in individuals under the age of 50 years [2]. Colon polyps, long-term ulcerative colitis and hereditary are causes of CRC and subjects are more often diagnosed between 65–75 years of age. Colon polyps are of three types, namely, adenomatous (a pre-cancerous condition), inflammatory polyps, and hyper-plastic polyps (cancerous condition). Polyps are larger than one centimeter in size or the presence of two or more polyps leads to CRC [3,4]. Dysplasia is another type of adenomatous polyp cell that is not cancerous cells, but is responsible for CRC. Polyps become cancerous while spreading to the blood or lymph vessels and then to the lymph nodes or other parts of the organs.

Curcumin (CMN) is a yellow hydrophobic polyphenol pigment extracted from the spice turmeric (*Curcuma longa*) that has been used for a long time in traditional Indian and Chinese medicine. It is a SERCA (sarcoplasmic/endoplasmic reticulum Ca^2+^-ATPase) inhibitor identified as having anti-cancer, antibacterial, anti-inflammatory, and antiprotozoal properties. Similarly, to other SERCA inhibitors, CMN exerts anti-cancer activity which has been shown to promote cell apoptosis derived from various tumors, for example, lung cancer, breast cancer, colon and ovarian cancer, and liposarcoma [5]. The CMN activates cell cycle arrest and apoptosis by evoking an increase in the concentration of Ca^2+^ in cancer cells regardless of normal cells. It was also reported that it induces endoplasmic reticulum (ER) stress and unfolded protein response (UPR) through SERCA2 inhibition in cells from thyroid cancer and liposarcoma, followed by caspase-3 and caspase-8 cascade-dependent apoptosis [6]. CMN is classified under biopharmaceutics classification system (BCS) class-IV for the reason of poor water solubility (>0.1mg/mL ≈11 ng/mL at 25 °C in aqueous buffer pH) and its low permeability through intestinal epithelial cells [7]. The rationale for the selection of drug in this study was: CMN is a natural substance, containing three acidic protons, it does not ionize over a wide range of pH. Additionally, high lipophilicity (log p = 3.29) will not hamper permeation. Many toxicological studies on this drug indicate that CMN is non-toxic even at high doses. The clinical safety, well-characterized pharmacodynamic profile, and low cost of CMN make it an ideal drug candidate for the development as an anticancer agent [8].

From a pharmaceutical background, it is well known that weakly acidic or basic and poorly aqueous soluble drugs are certainly devoid of dissolution and bioavailability. In this case, converting the crystalline form of a drug into amorphous is a dynamic approach to improve the bioavailability, solubility, and dissolution. The extreme crystalline nature of the drug requires additional energy for the dissolution medium to interrupt the disordered amorphous form because crystalline lattices are highly ordered [9]. Though the demerits of amorphous drugs are likely to recrystallize and become thermodynamically unstable. Therefore, the most challenging task of modern drug development in the pharmaceutical industry is to improve the solubility of insoluble or poorly water-soluble drugs [10].

There are numerous physicochemical processes to enhance the solubility such as the increase in surface area by micronization, nanosizing, conversion of amorphous form, enhancement of wettability, prodrugs and salts approach, liposomal delivery, interaction with a hydrophilic carrier, and solid dispersion (SdD) technique. SC has wide-open applications as one of the most well-known delivery approaches to enhance the solubility and dissolution of drugs [9,11]. 

Briefly, SdD is defined as a molecularly dispersed blend of drugs with an inert carrier, i.e., hydrophilic carrier. A conversation of crystalline into an amorphous form of any drug with a reduced particle size for improving the wettability is the principal approach, wherein, SC could improve the solubility and dissolution of drugs [9]. Perhaps, the presence of a polymer in the SdD complex can possibly increase the stabilization, thus preventing the drug from recrystallization. These mechanisms responsible for the SdD properties consist of drug–polymer interactions such as anti-plasticization, the glass transition temperature, and polymeric carrier viscosity [12,13]. This method is mainly suitable for drugs in the thermolabile category (low melting points). 

The method of solubilization includes breaking the ionic or intermolecular bonds in the solute, the separation of particles to create space in the solvent for the solute, and enhancing interaction between the solvent and the solute ion or molecule. The lipophilic or poorly water-soluble drug dissolves in an inert carrier, which enhances the solubility of the drug, thereby improving oral bioavailability. The drug is molecularly dispersed into the carrier matrix as very fine particles [14], thus improving the dissolution rate in the carrier of the solid solution. The solid solutions can be categorized based on the miscibility (continuous vs. discontinuous solid solutions) and the molecular distribution of drugs present in the carrier (amorphous, substitutional, or interstitial). 

Typically, HME technology is frequently used for the preparation of SC, especially BCS class II, III, and IV drugs, i.e., low solubility and low permeation. HME technology is a continuous process that functions in particular conditions that could be optimized for the solubility enhancement, i.e., rate of the feed materials, rotating screw speed, shear stress, and temperature. Mixing and conveying are achieved by pouring raw materials in the hopper-fed via a heated barrel, which contains a rotating screw that splits into different regions [15]. Raw ingredients are fed via a heated barrel, which contains rotating screws, which are distributed into two sections of mixing and conveying. At this point, contents start to melt first, followed by a plasticization process and travels at the entire length of the screw barrel [16]. The rationale of HME is the process of applying heat and pressure to melt a polymer and forcing it through an orifice in a continuous process; hence, HM stands to apply heat transfer only. Advantages of HME technology are a continuous process and modifying or controlling the release of the drug from the reservoir, short processing time, easy to scale up, does not require any special solvents. Hence, the obtained final products improve the bioavailability and dissolution due to the uniform dispersion of drugs and carriers. In addition to SC, the HME technology effectively alters the pH by utilizing the pH modifier. This is a potential technique to control the rate of release of lipid-soluble drugs (i.e., curcumin) to enhance the rate of bioavailability [17]. 

Polyethylene glycol (PEG) is a high molecular weight polymer of ethylene oxide [18]. PEGs are also soluble in many polar organic solvents such as acetone and alcohols. PEGs are hygroscopic, which means they attract and retain moisture from the atmosphere. It is a potent hygroscopic substance that decreases with increasing molecular weight and PEGs behave as per the Newtonian flow because of the kinematic viscosity, i.e., increases as temperature decreases.

This investigation is advanced in pharmaceutical research, which includes two methods, HM and HME, in developing SC that effectively enhances the dissolution and bioavailability of CMN. The objectives of this research work are the optimization of variables in the preparation of SC, and to characterize the aqueous solubility, dissolution of drug release, cytotoxicity and in vivo toxicity.

## 2. Results and Discussion

### 2.1. PS Studies

This study provides essential data on the impact of the different carriers on CMN solubility. A typical linear curve is obtained in the concentration order from 3.74 × 10^−4^ to 4.61 × 10^−4^ Mm and 6.27 × 10^−5^ Mm to 6.78 × 10^−4^ Mm from 25 and 37 °C, respectively. These results reveal as A_L_ type of phase solubility profile (the drug solubility constantly increases as a function of carrier concentration) caused by changes in the forces of interaction, such as hydrophobic forces and Vander Waals force between the drug and the carrier (as shown in Table 1). The obtained slope from the PS diagram is to determined as less than 1 in PEG indicating the complex was 1:1 stoichiometry [19]. PEG 6000 and PEG 4000 showed the ideal complexation constant in the range of 100 to 1000 M^−1^. The stability constant (K_1:1_) estimates from the slope and intrinsic intercept values of the solubility curves. The PS diagram was obtained (as shown in Figure 1a) by plotting the percentage of CMN dissolved against the carrier concentration (% *w*/*v*). (Table 1).

The determined value of ΔG was negative in PEG 6000 and PEG 4000, showing the spontaneity of binding (solubilization) and the binding decreases with an increase in the PEG molecular weight. The calculated value of ΔH observes to be positive (endothermic) in all the PEGs. Moreover, the ΔS value in PEG 6000 and PEG 4000 is detected to be high (8 and 9 KJ.mol.K^−1^), demonstrating that the complex produced an endothermic effect, which was reported by Mahmoud [20].

The ideal interaction constant observes at PEG 6000 (309 and 377 M^−1^) and moderate interaction of PEG 4000 (141 and 176 M^−1^) at 25 and 37 °C, respectively, due to additional contacts offering intermolecular interactions between solvent molecules. This is expected to control the interaction of the solvent with the heteroatom [20]. The dielectric constant of an aqueous solution of PEG decreases with the increase in concentration; apart from the high solubility of CMN in PEGs (glycol schemes), the hydrophobic interaction (hydrogen bond) plays a much essential part in the solubility of the CMN in the long non-polar part of PEGs. Subsequently, PEG 4000 is also shown as a desired stable complex, but it is slightly lesser than PEG 6000 because of the low viscosity of PEG 4000. 

### 2.2. In Silico Interaction

Computer modeling of PEG controls the selective points of attention of their molecular structure. The in silico interaction study was established, and it was easy to understand the stability of the complex containing drug and ligand molecules. The perfect molecular modeling for the 1:1 complex of CMN:PEG is shown in Figure 1b. PEG contains the major ketone group and the hydroxy hydrogen interaction with the ethylene hydrogen region of the CMN comprising the benzene ring and the methoxy CH_3_ hydrogen-oxygen group in the distance range of 2.7 Å [21]. In addition, the two hydrophilic chains of polyoxyethylene (poly(ethylene oxide) are involved in the interaction with CMN containing ketone group at a bond distance of 1.9–2.5 Å. The ideal interaction of hydrogen between the CMN/PEG complexes includes in the broad interaction with CMN containing ketone group. A close-fitting CMN/PEG appears to clarify the tiny variations in enthalpy and positive changes in entropy observes in the complex formation, resulting from enthalpy–entropy compensation [22]. Overall, the in silico interaction study shows that the nonionic PEG comprises a better non-bonded interaction with CMN. These studies provide a better understanding of complexation with the PEG and the results match exactly with the data obtained from PS studies. The overall view on the aqueous solubility of CMN in the SC is found to increase the carrier concentration.

### 2.3. Solaq

The Solaq of pure CMN in double distilled water at 37 °C is 0.004 mg/mL for 24 h. In PEG 6000 and PEG 4000 (1:3 to 1:7), the solubility of CMN enhances by ~180 to 220 and ~150 to 180 from HME technology and ~160 to 200 and ~140 to 170-fold form HM method, respectively (Figure 2a). The Solaq is because of greater hydrogen bonding in the water particles to an electron-rich oxygen atom in the polymer chains of PEGs. Surface properties enhance the wettability by reducing the surface tension of the vehicle employed so that the drug molecules easily penetrate into an aqueous environment. 

### 2.4. Drug Release

The curves of mean dissolution of pure CMN, HME- and HM-SCs is shown in Figure 2b. It appears that the rate of release (slow-release phase) of pure CMN was found to be 1.62% toward the end of 30 min due to its high hydrophobicity that influences the powder to float on the dissolution medium and anticipates contact with the bulk of the dissolution medium (double distilled water). The dissolution on SC with their carriers exhibits an initial high burst release (48.73–83.28%) in the initial 5–6 min showing the complex develops with the carriers or converts into amorphous/crystal lattice [23]. After 5 min, the drug releases from PM and SC in a low quantity with steady-state pattern.

The PEG 6000 SC from HME and HM complex, shows 91.62 and 78.18% of the CMN was released at the end of 30 min with the initial burst release of 83.28 and 73.39% respectively. On the another hand, PEG 4000 complex of HME and HM releases found to be 71.32 and 62.68% at 5 min and by then, 73.24 and 64.02% at 30 min, respectively (Figure 2b).

Li et al. (2015) studied that the release pattern due to metastable supersaturation of CMN at a PEG concentration moistened with proper dissolution medium [24]. Other than non-ionic characters, viscosity places an important role in the drug dissolution by improving the interaction of the CMN with the dissolution medium while the drug fully dispersed/dissolved in the molten carrier Ahuja et al. (2007) reported that the exchange of the glassy matrix system to an amorphous form of the drug and the quick solidification of the PEG [25]. Hydrophilic PEG carriers produce a successful conversion of CMN in the amorphous form to improve the drug release in HME technology. However, the result from HM-SCs likely improved the drug release owing to the partial conversion amorphous form of the drug.

### 2.5. Solid State Characteristics

#### 2.5.1. FTIR

The FTIR spectrum of the drug, PEGs, PM, and SCs is shown in Figure 3a, where the CMN reveals the number of characteristic bands representing O-H stretching (alcohol, 3324.68 and 3015.16 cm^−1^), C=O stretching (1742.37 cm^−1^), C=C stretching (alkenes, 1629.55 cm^−1^ and aromatic, 1597.73 and 1507.1 cm^−1^), C-H bending (1426,1, 1371.14, 960.377 and 810.92 cm^−1^) and stretching vibration which was snug fit the earlier report [26].

The characteristic bands of both PEG 6000 and PEG 4000 demonstrate that O-H extending vibration (3479.92 and 3438.46 cm^−1^) PEGs vanished, and C=O extending vibration (1642.09 and 1645.95 cm^−1^) attributable to CMN was discovered to reduce wavelength with PM and SC, suggesting that the complex was developed by hydrogen bonding. Similarly, Thorat et al. (2015) reported that the single peak seen at 2876.31 and 2874.24 cm^−1^ (C-H extending) might be the complexing option site of the drug with the carrier (Figure 3a). Likewise, SC demonstrates the nonappearance of the N-H extending band at 3343 cm^−1^, proposing the intermolecular hydrogen holding between the drug and the carrier [27].

#### 2.5.2. PXRD

The CMN exhibits sharp peaks XRD patterns (due to the occurrence of a crystalline drug form) at 2θ of 8.98°, 17.38° and a series of tiny peaks at 23.48, 24.72, 25.68, 26.22, and 27.5° (Figure 3b). The carrier typical peaks such as PEG 6000 (2θ of 19.38, 23.54, and 27.08°) and PEG 4000 (2θ of 19.46, 23.62 and 27.28°) indicate the crystalline domain within the amorphous polymeric material [28]. 

The obtained outcomes in PEG reveal that the amorphous nature is vital in both PM because of the reduced intensity and a considerable amount of CMN dissolves in a solid PEG matrix in its amorphous structure was assured by various characteristic peaks of SCs signifying the transformation of a crystalline form of CMN to an amorphous form of SC. The indistinct idea of the carriers was set up at SC completely in their individual diffractograms (as shown in Figure 3b).

#### 2.5.3. TGA 

The CMN, PEGs, and PM and SCs were investigated by TG analysis and the thermogram is shown in Figure 3c. The drug starts to lose mass at 101 °C, and absolute mass loss identifies at 530 °C. From the thermogram, three zones of mass loss are detected with PEGs—weight reduction in the range of (i) 40–140, (ii) 250–465, and (iii) finish weight reduction observes over 523 °C. Subsequently, the PM of PEGs dissipated between 270 and 532 °C. Additionally, the SC from PEGs exhibits mass loss from 290 to 535 °C. The thermal treatment of the difference conveys that the distinctions in a hot solution of CMN, PM, and SCs are insignificant and we could confirm that there were no compound interactions between the SC complex. These alterations among the PM and the SC within this phase (loss of mass) may be a host–guest interaction that could limit the development of the drug molecules. The physicochemical properties, such as, liquefying, boiling, and sublimation points are altered when the visitor molecules are transferred through the host molecules in PM. In a complex formation of PEGs, the absolute absence of the endothermic peak linked to CMN could be the arrival of water molecules or the entire conversion to the amorphous frame or dissolution of the crystalline form into the liquid carrier. The distinctions in the thermal decomposition of PM and SCs are insignificant; we could confirm according to the report represented by Pramono et al. (2016), no substance interacts with the complex [29].

#### 2.5.4. DSC 

The DSC curve of CMN (179.8 °C), PEG 6000 (61.1 °C), and PEG 4000 (59.7 °C) exhibited endothermic peaks, which conforms to previously reported data as shown in Figure 3d. In the thermogram of PM, the physiochemical properties, such as sublimation, melting, and boiling points subsequently changed while the CMN interacted with PEG.

In SC of PEGs, the CMN endothermic peak was completely absent due to the water molecule being liberated or completely converted to an amorphous form or dissolution of the crystalline CMN into the liquefied PEGs. A tiny peak near 208 °C observed in the SC from HM with reduced intensity may relate to the melting of CMN at higher temperatures and indicates a reduction in the crystal nature of the drug [25].

### 2.6. CCD Outcomes

From the initial studies, SC prepared with PEG 6000 binary complex by HME technology exhibited enhanced solubility and release rate of CMN than pure CMN and HM-SC. The CCD is a dynamic second-order experimental design associated with 13 experimental trials (as shown in Table 2). The concentration of PEG 6000(X_1_) and the screw speed (X_2_) were investigated as independent variables at two-factor, three-level and the critical quality attributes selected were Y_1_; aqueous solubility (Solaq; mg/mL) and Y_2_; release at 5min (Rel_5min_; %) as responses.

The coded values of independent variables concentration of PEG 6000 (CP) (−1; 300 mg) low value and (+1; 700 mg) high value and screw speed (SS) (−1; 80 rpm) low value and (+1; 120 rpm) high value was chosen based on the results from earliest experimental trials.

#### 2.6.1. Sloaq

The model (quadratic) is significantly suitable for the dependent variable as mentioned by the RSM-CCD. The polynomial equation for response (aqueous solubility; Y_1_) represents the following Equation (5),
Y1 = −1.75378 + 0.002240 × A + 0.038265 × B − 0.00625 × A × B− 0.001 × A2 − 0.000175 × B2 (1)

The sequential lack-of-fit test, F-test, and analysis of variance (ANOVA) results were analyzed to check the model’s ability. The quadratic predictive model is significant and valid for the response Y1 (*p* = 0. 0001). An R^2^ value of 0.9916 for Y_1_ (Solaq) indicates a good correlation between the experimental and predicted responses (0.9471) [30]. The reasonably minor estimation of CV and adj. R^2^ (1.73 and 0.9856, respectively) signifies a high grade of precision and reliability of the practical estimation and a large possible correspondence among the experimental and predicted values (Table 2). The rest of the factors and their collaborations were not prominent (*p* > 0.05). 

The interaction effects of the factors on response were demonstrated by graphical illustration, which depends on the CC (X_1_) and SS (X_2_) and their Solaq of CMN (Y_1_) as shown in Figure 4(ai). An increase in the amount of PEG 6000 from 300 to 700 mg resulted in a significantly improved solubility of CMN from 0.612 to 0.981 mg/mL^−1^. This report shown as the PEG 6000 can be efficiently liable for the Solaq of the CMN. The SS showed an increase (80 to 120) in rpm on increasing the solubility is significant.

#### 2.6.2. Rel_5min_

The following polynomial model for response (release at 5 min; Y_2_) is represented by the following Equation (6). It is evident that all the three independent variables, i.e., CP and SS, have positive effects on the response by Equation (6).
Y2 = +81.66 + 2.96 × A + 0.6012 × B (2)

All experiments of independent factors such as CP and SS estimate to have significantly progressive effect on the Rel_5min_ from 74.11 to 91.47% and is obvious from the positive value for its coefficient presented the release improved with increasing CP and SS. 

The positive effect of SS was established to be less than that of CP. The interaction among the independent factors is also determined to be significant [31]. In general, the obtained model is significant (F-value = 165.74; *p* < 0.0001) while the lack of fit is not significant (F value of 0.1344; *p* = 7.92). The values for predicted (0.9471) and adjusted (0.9856) R^2^ values are in possible evident. The concentration of PEG 6000 significantly enhances the release of the drug from SC at the finish of 5 min. If low PEG 6000 concentration, release from SC could not be significant, when it is up to 700 mg, the maximum release could be achieved up to 91.47% (as shown in Figure 4(aii)). It shows the release rate of the drug increased quickly by increasing the SS. These effects were significantly shown for CP of 300 to 700 mg, and the center point of SS reported that the release of the drug significantly increases as the SS was increased, which could be due to absolutely amorphization of the drug.

The final desired solution given by software was utilized to prepare an optimized (P14) formulation. To optimize the method validation, response variables (Solaq and Rel_5min_) were set at a maximum range and the optimized formulation generated by design was found to be 100 mg of CMN, 450 mg of PEG 6000, and 85 rpm SS for the preparation of HME-SC. Experimental value of Solaq (0.852 ± 0.02 mg.mL^−1^) and Rel_5min_ (91.87 ± 0.208%) was compared with that of the prediction value of 0.8396 mg.mL^−1^ and 93.35%, respectively. From the findings, the prediction error as found to be 0.0125 and 1.48% (Figure 4(bi,bii)) and the negligible error percentage indicates that the developed model is ideally good and the predicted results are in argument with the experimental data [32]. The optimized HME-SC complex carried out for further studies.

### 2.7. SEM

The SEM image in Figure 5a supports the surface morphologies of pure CMN (i), PEG 6000 (ii), and HM-SC (1:5) (iii) and optimized HME-SC (iv). The pure CMN looks like a crystalline structure of characteristic prism-textured with mean particle size shown as 15 µm and PEG 6000 seemed to be smooth surface particles with a crystalline-amorphous surface (stereo character). 

In the topological changes supposed in the drug particles of the HM-SC partially converted into it amorphous, which was revealed by release study and FT-IR study. Hence, optimized HME-SC; the drug surface seems to be more porous and was observed to be identical and homogeneously dispersed (CMN and PEG 6000 morphology had vanished) at the molecular level. The SEM image clearly revealed the crystalline CMN was absolutely converted into an amorphous form.

### 2.8. DLS

The hydrodynamic PS of HM-PM was found to be 235 ± 7.2 nm with a PDI of 0.286 (as shown in Figure 5b) revealed as a bi-modal size distribution (complex containing at least two molecules) in the solution of the PM. The hydrodynamic diameter of HME-SC was 144.6 ± 5.6 nm with a PDI of 0.236 (Figure 5(bii)). Kumar et al. (2015) demonstrated that the ZP is an important parameter to study the nature of the particle surface and predict the stability of the complex (long-term stability) [33]. The ZP of HM-SC and HME-SC was found to be −26.1 and −29.3 mV (as shown in Figure 5(biii,biv)), respectively, and it shows that the surface charge was reduced to some extent. The report exposed a low PDI indicating the uniformity of particle size distribution and HME-SC had a capable of adequate repulsive force to inhibit settling or aggregation/agglomeration while long-term storage due to the CMN completely converted to amorphous form.

### 2.9. Dyeing Effect 

As shown in Figure 6a, pure CMN, HM-SC (1:5) and HME-SC (1:5), while adding 10 mg of pure CMN in 15 mL of distilled water, the drug floated to the surface owing to its lipophilic nature. However, the filtrate is clear and is transparent with HME-SC. Subsequently, HM-SC shows as the yellow color solution when it was added to the same volume of water (Figure 6(ai,aii,aiii)). The dyeing effects indicate that the solubilizing capacity is predominant in HME-SC [30]. Xu et al. (2015) studied that this direct functionalized dye test without assistants demonstrated that HME-SC showed better coloring impact (as shown in Figure 6(bii)), signifying that the solute solubility of complex expanded all the more equally in water after incorporation of these complex [34].

### 2.10. MTT Assay

The cytotoxic outcomes of pure drug and HME-SC on SW480 (Figure 7a) and Caco-2 (Figure 7b) cells lines are presented from the MTT assay. The IC_50_ value for SC was found to be 72 (SW480) and 40 µM/mL^−1^ (Caco-2), while that of pure CMN ranged from 146 to116 µM/mL after 24 h treatment.

The results from the MTT assay reveal better cytotoxic activity than pure CMN. Manju and Sreenivasan reported that these outcomes can be attributed to the influence in the deviations of cellular uptake profile leading to the potential effect of SC [35,36]. The outcomes of this study proved that the SC could deliver the CMN to SW480 and Caco-2 cells by active targeting by the progression of endocytic with potential cytotoxicity and aqueous solubility of HME-SC. The cell viability of both pure CMN and HME-SC decreases with increasing the concentration, however, SC was significantly potential than that of native CMN. The MTT assay was performed for 24 h. At 24 h, 0.2 mL of 100% DMSO is sufficient to solubilize the drug from the complex after incubation and can inhibit the human adenocarcinoma cells (SW480 and Caco-2). On the contrary, in vitro conditions are different from an in vivo physiological environment.

### 2.11. AO/EB Staining

The morphology of control or live cells that look like a bright green color and have identical chromatin with an integral cell membrane is clearly shown that they do not undergo apoptotic deviations as depicted in Figure 8. On the contrary, more exploited cell death was observed on the stained cells treated with HME-SC, i.e., apoptotic cells and necrotic types of cells (Figure 8c). Abel et al. (2018) examined that the morphological modifications during apoptosis are necessary benchmarks in cell death that can be observed by AO/EB staining [37]. If the cells faced a particular cell death pattern, the cytology changes would have been investigated by EB/AO staining. As specified by the emission of fluorescence and chromatin geologies; the cells can be categorized as the following types: viable cells—these cells have uniform and sorted structures with green fluorescing cores (Figure 8(ai,bi)); early apoptotic cell—these have intact membranes with green fluorescing cores, yet DNA fragmentation was seen to have started and chromatin buildup was perinuclear and visible in green splendid or patches (Figure 8(aii,bii) and 8(aiii,biii)); late apoptotic cells—these had fused or divided chromatin with orange to red fluorescing nuclei; and necrotic type of cells—these have considerable or swollen structures and invariably orange to red fluorescing nuclei with the presence of chromatin cleavage (Figure 8(aii,bii) and 8(aiii,biii)) [38]. These outcomes indicate that the HME-SC complex treated could lead to cell death through apoptosis and little necrosis.

### 2.12. Hoechst 33528 Staining

The cytological changes were observed after 24 h incubation from SW480 and Caco-2 cells, physical counting of live, apoptotic, and necrotic cells rates were measured by blue Hoechst staining of SW480 and Caco-2 cells and is shown in Figure 9. From this finding, more amounts of apoptotic cells are observed in both SW480 and Caco-2 cells with the least number of necrotic cells (Figure 9c) when treated HME-SC than pure CMN. Vignesh et al. (2014) reported that a feature of Hoechst staining shows to detect the changes in cell cytology, with an exclusive pattern of cytoplasm and cell nuclei at significant level diagnosing the cell death [39]. The HME-SC-treated cells were clearly observed that the early apoptotic summary, i.e., cell shrinkage, enlarged chromatin, and breaking of cells and nucleus and little numbers of necrotic cells were also observed.

### 2.13. Toxicity of HME-SC in Zebra Fish Embryos

Kim et al. (2013) recommend zebra fish embryos as being a suitable model for novel toxicity research [40]. The hatched embryos (after 96 h) are considered to have survived and the percentage of embryos that died by 96 h represents the mortality rate; these were compared against the levels in the control group. Figure 10b shows the levels of mortality and survival embryos exposed to different concentrations (25–300 µg/mL) show no significant toxicity (Figure 10a). The HME-SC-treated embryos were observed under compound stereo microscope to detect the prevalence of abnormalities/malformations, such as pigmentation, bend of the tail, pericardial edema, non-depleted yolk, malformed spine, and deformity of the pericardial sac of the larvae [41]. The HME-SC treated zebrafish egg embryos did not exhibit any major disruption in their normal development along with the percentage viability of larvae not producing any significant toxicity (Figure 10b). Therefore, the zebrafish toxicity study strongly suggests that HME-SC could be a worthy preparation for a novel drug delivery system and biomedical engineering.

## 3. Materials and Methods

### 3.1. Materials, Cell Lines and Regents

The list of chemicals that were purchased include: curcumin (CMN; 99% of purity; SRL Pvt. Ltd, Maharashtra, India) and Polyethylene glycol (PEG; S.D. Fine Chem. Pvt. Ltd, Mumbai, India). Cell Culture: The colorectal adenocarcinoma cell lines (human) of SW480 and Caco-2 were purchased from the NCCS (National Center for Cell Science) in Pune. In DMEM the cells were cultured (Sigma-Aldrich, St. Louis, MO, USA), supplemented with 10% fetal bovine serum (FBS—research grade; Sigma, Madison, WI, USA) in 96-well culture plates and 1% of streptomycin (100 µg)/mL/penicillin (100 U/mL) were used as an antibiotic (Hi-media, Mumbai, India), at 37 °C in a humid atmosphere of 5% CO_2_ in a CO_2_ incubator (Thermo Scientific, Waltham, MA, USA). Used reagents and chemicals were of analytics grades. 

### 3.2. Phase Solubility (PS) Studies

The PS study was carried out using Higuchi and Connors with slight modifications [42]. Briefly, this study was carried out by adding an excess amount of CMN to the 25 mL aqueous solutions of various concentrations (1 to 15%) to the PEGs. Eppendorf tubes containing the content set were placed in a water bath at a uniform temperature (at 25 and 37 ± 1 °C) for 24 h until the point reached equilibrium and shaken regularly for 30 min intervals. Hence, the content was filtered by the 0.45 µm Millipore membrane filter, diluted appropriately, and the absorbance was noted at λ_max_ 425 nm with a UV-spectrophotometer (Agilent Cary 60, Santa Clara, Stevens Creek Blvd., USA). The PS curve can be constructed by the obtained slope and intercept estimated for stability complexation constant (K_1:1_); hence, intrinsic solubility of the CMN is directly proportional to the intercept (Equation (1)).
K(_1:1_) = X/((Y (1−X)(3)
where, X and Y refer to the slope and intercept, respectively; moreover, the enthalpy change (ΔH) was determined by the following Equation (2)
ln K_2_/K_1_ = ΔH ((T_2_ − T_1_))/kRT_1_T_2_(4)
where K_1_, K_2_, and T_1_ and T_2_ refer to the stability constants and different temperatures in Kelvin of 25 and 37 °C, respectively [43]. The change in entropy (ΔS) and Gibbs free energy (ΔG) upon complexation/solubilization were computed from the following Equations (3) and (4), respectively.
ΔG =RTlnK (5)
where R= −8.314 mol.J^−1^ and K is gas constant
ΔS = ΔH ((ΔH − ΔG))/((ΔG)) (6)

### 3.3. Molecular Modelling 

The molecular interaction studies were supported by the BIOVIA discovery studio 2017 (DS) platform using drugs and carriers. The structures were collected from the PubChem and ChemBook database and converted into a PDF file then the binding region specified in a sphere based on the functional group present in the carrier [44]. Primarily, the grid was created close to the carriers of PEG (CAS No.: 25322-68-3) coordinates of X (10.521), Y (−2.624) and Z (−0.861). The final ten best conformations were chosen to study complexation among the CMN and PEG. The PEG is docked around the CMN and close to the ideal composite would save for the interaction study using the C-Docker protocol in DS. CHARMm established algorithm was utilized to construct the collaboration among drug and carrier complexes.

### 3.4. Preparation of Physical Mixture (PM) 

The PM of the varying compositions of CMN and PEGs (1:3 to 1:7) was physically mixed by pestle and mortar as specified by the guidelines of geometrical mixing (100 mg of drug used in each preparation), followed by screening (# 120; 150–125 µm). 

#### 3.4.1. Preparation of Soluble Curcumin (SC) by Melt Casting (HM) Method

The SC was prepared by adding CMN to the liquefied PEGs at 70 °C with continuous stirring at 700 rpm for 15 min until obtaining a homogeneous dispersion. The liquefied solution was immediately cooled at room temperature (25 °C), powdered, sieved, and stored at 27 °C in a desiccator [44].

#### 3.4.2. Preparation of Soluble Curcumin (SC) by Hot-Melt Extrusion (HM) Method

The CMN (10% *w/w*) and PEG 6000/PEG 4000 (30 – 70% *w/w*) were mixed and milled thoroughly using a mortar and pestle and these PMs were added to the twin-screw extruder (CHT20-B, Twin-Screw Extruder, Nanjing, China) at 90 °C using a specification of standard configuration as described by the supplier [3]. The speed of the screw was adjusted from 80–120 rpm. Then, respectively obtained extruded was pulverized into a powder particle using a pestle and mortar and sieved through #120 mesh with a particle size of 150–125 µm.

### 3.5. Aqueous Solubility (Solaq) Study

An excess of samples (CMN, PM, and SC) with double distilled water were taken in a 50 mL Eppendorf tube and placed on a constant water bath, and temperature was maintained at 37 ± 0.5 °C for 24 h and shaken at 30 min intervals [45]. Subsequently, the contents were filtered through a 0.45 µm Millipore membrane filter, diluted suitably, and the UV absorbance was measured.

### 3.6. Dissolution Study

The dissolution study was performed in 900 mL Millipore MilliQ water at 37 ± 0.5 °C and 50 rpm by a dissolution apparatus as per Indian Pharmacopoeia specification (DS8000, Lab India, Maharashtra, India). At zero time interval, samples were placed into a jar with the above specification every 5 min until 30 min, 5 mL of samples were pumped out and filtered using Whatman filter paper (11 µm). The CMN dissolved in media was determined by a UV-spectrophotometer. A correction represented for the fresh dilution initiated by replacement of the samples with a Millipore MilliQ water to maintain the sink condition [43].

### 3.7. Solid State Characteristics 

#### 3.7.1. Fourier Transform Infrared Spectrophotometric Analysis (FT-IR)

FT-IR investigation was carried out to assess possible interaction (difference in structure) between the CMN and excipients. IR spectrum of the solid samples was analyzed in the solid powder by the KBr disc method in the wavenumber range of 4000–400 cm^−1^ with a scan speed of 1 cm^−1^ in an FT-IR spectrophotometer (JASCO/FT-IR-6300, Tokyo, Japan).

#### 3.7.2. Powder X-ray Diffraction Analysis (PXRD)

The PXRD patterns of samples were observed with Rigaku Ultima III XRD (Rigaku Co., Ltd., Tokyo, Japan). PXRD was performed through a Kb filter and Cu Energy at a current of 30 kV and a 15 mA current. The samples were reliably spun and examined at a degree of 1 °/min over a 2θ range of 5– 80° (data shown at 5– 40°) in a pre-stacked PC program. 

#### 3.7.3. Thermogravimetric Analysis (TGA)

The powder sample heated in the range of 20 °C/min from 30 to 700 °C with a nitrogen atmosphere (flow rate 10 mL/min), in open aluminum pans containing 3 mg of the sample using TGA 4000 (Perkin Elmer, Akron, OH, USA).

#### 3.7.4. Differential Scanning Calorimetric (DSC) Analysis

The DSC thermogram of powdered samples was detected with a thermal analysis system (DSC; Pyris 6, Diamond TG/DTA; Perkin-Elmer Instruments, Shelton, CT, USA) with an underflow (20 mL/min) of nitrogen with 3 mg samples positioned in a sealed aluminum pan. The sample was heated in the range of 10 °C/min from 20 to 300 °C. The enthalpy changes were calculated with the pure drug and complexes [43].

### 3.8. Optimization of Ideal Soluble SC Complex 

The dynamic Central Composite Design (CCD) chosen for the less trial preliminaries with second-order trial configurations The Design-Expert software® version 11 (Stat-Ease Inc., Minneapolis, MN, USA) was used for arithmetical assessment by ANOVA, model equations construction and response plots in 3D for every response. The amount of carrier and speed of the screw (process variables) are independent and the resultant dependent variables are aqueous solubility and time of dissolution that affect the preparation of a better soluble SC complex [46]. 

### 3.9. Particle Size (PS) Analysis

The determination of mean particle size, polydispersibility index (PDI), and zeta potential (ZP) of drug and SCs was measured through dynamic dispersion of the light technique using the Zetasizer (Malvern Instruments Ltd., Nano ZS90, Malvern, UK) [47]. 

### 3.10. Morphology

A thin smear of the samples was gold-coated (100 Å) via a sputter coater prior to the amplifications at a voltage of 5.0 kV observed. Then, according to Carl Zeiss Microscopy Ltd, UK, scanning electron microscope (SEM), photographs were carried out by a scanning electron microscope of zig-zag pattern that were carried out at a voltage of 5 KV acceleration [48].

### 3.11. Dyeing Experiment

A simple novel test facilitates the efficient solubility of a drug (usually colored) in an aqueous medium. Briefly, 10 mg of pure CMN and equivalent wt. of SCs were dissolved in distilled water (15 mL) followed by sonication (5 min) and filtration [49]. White linen (cotton) material pieces of similar size (8 × 4.5 cm^2^) were soaked in the above solution diluted to 50 mL and dried for 1.5 h. The photograph of both solutions and dried clothes were captured.

### 3.12. MTT Assay 

MTT (3-(4,5-Dimethylthiazol-2-yl)-2,5-diphenyltetrazolium bromide) assay applied on the samples using SW480 and Caco-2 cell lines (cell passage number was 3). Briefly, cell passaging was performed when the cell line reached the growth point of the culture, it shelters most of the bottom of the culture vessel, or approximately 90% confluence. Cells must be resuspended, washed, used in experiments, frozen for later use, or re-seeded for further development of new culture vessels. The cells were cultured into 96-well plates and seeded at a density of 5 × 10^3^ cells/mL in 0.2 mL/well and 100% DMSO (Sigma-Aldrich) solvent was used as a control (0.02%). After 24 h incubation, the prepared various concentrations (20–200 µM/mL) of SCs were treated with culture and the MTT reagent (5 mg/mL in PBS; 20 µL/well) was added to the medium and incubated again for 4 h at 37 °C. The obtained purple formazan product of dissolved by the addition of DMSO solvent (0.1 mL) to all the wells [50]. The above solution absorption was measured at 570 nm by means of plate reader (iMark, Bio-Rad, Hercules, California, USA). To determine the IC_50_ concentration, SC complex concentration is necessary to reduce the absorbance to half of the control. Data were collected in triplicates, three times each, and three mean values were used to calculate (OriginPro 2018, Northampton, MA, USA) the mean ± SD. Inhibition rates were calculated from these data using the formula:

Percentage inhibition = Mean OD of cell in control − Mean OD of cells treated HME-SC/Mean OD of cell in control × 100

### 3.13. Apoptosis Study

#### 3.13.1. Acridine Orange and Ethidium Bromide (AO/EB) Twin Staining

Apoptotic morphology investigation was carried out by using acridine orange and ethidium bromide (AO/EB) twin staining method [51]. Briefly, determined IC_50_ concentration of the SC complex was treated with SW480 and Caco-2 cells (cell passage number was 8) and incubated for 24 h followed by washed with ice-cold phosphate-buffered saline (PBS). The cell cluster was was collected and diluted with PBS and the cell density was maintained at 5 × 10^5^ cells/mL. The AO/EB twin stain was treated with the above solution with the specification of 2.5 µM of EB and 3.8 µM of AO in PBS on a clean mounted glass slide. A Carl Zeiss fluorescent microscope (Axioscope 2plus, Selm, Germany; 400× magnification; Axicam ERc 5c camera; N.A. of 0.65) with 450–490 nm UV filter was used to analyze the morphology of cells within 15 min. Approximately 300 cells/sample were estimated as live, necrotic, and dead cells including structure nucleus, membrane integrity and percentage calculated. Data were collected in triplicates, three times each, and three mean values were used to calculate (OriginPro 2018, Northampton, MA, USA) the mean ± SD. 

#### 3.13.2. Hoechst Staining

The Sw480 and Caco-2 cells (cell passage number was 8) were cultured in separate 6-well plates and treated with IC_50_ concentrations of samples. Control and treated cells were collected after 24 h of incubation and stained (Hoechst 33258 stain; mg/mL; aqueous) at room temperature for 5 min. The fluorescent microscope was fitted with a 377–355 nm filter, randomly observed 300 cells with 400X magnification an N.A. of 0.65. [52]. Data were collected in triplicates, and the mean values were used to calculate the average (OriginPro 2018, Northampton, MA, USA) the mean ± SD.

### 3.14. In Vivo Zebrafish Toxicity Study

Adult zebrafish (*Danio rerio*) were purchased from the Tiruchirappalli district of Tamil Nadu. Healthy eggs, embryos, and larvae were produced after 3 h of spawning. After 3 h of post fertilization (hpf), eggs were collected and ten healthy embryos were placed into a 24-well culture plate containing 1 mL of E3 media [53]. The control was maintained as a placebo and the samples (25–300 µg/mL) were treated for 96 h. The embryos were examined daily for their morphological changes, hatching, and survival rate, edema, spine dislocation, non-depleted yolk, tail bend as indicators of toxicity.

## 4. Conclusions

This study set the trend to examine the effect of HME technology applied to enhance the solubility and dissolution of the curcumin and evaluated the anticancer potency of soluble CMN on human colorectal adenocarcinoma cells. The phase solubility and molecular modeling study produced the complex stability which concluded the better SC complex. Subsequently, the ideal 1:1 stoichiometrically governed SC complex, further optimized by RSM-CCD, in the 1:5 ratio was shown as better for CMN solubility throughout. Enhanced curcumin solubility up to 220-fold higher than pure curcumin and the novel dyeing test was performed for the first time with the complex of CMN, signifying its solubility. The little IC_50_ concentration and in vivo toxicity study confirmed by zebrafish model established that the HME-SC is an ideal and potential preparation for the treatment of colorectal cancer. In this study, a useful approach was provided to obtain maximum soluble curcumin products. This research provided an existing and unique technique for the application of a valued cancer treatment.

## Figures and Tables

**Figure 1 pharmaceuticals-14-00939-f001:**
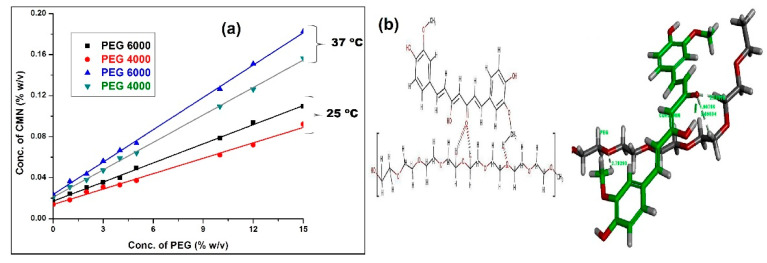
(**a**) Phase solubility curve of CMN in MilliQ water at 25 and 37 °C in the presence of PEG; (**b**) molecular modeling for the 1:1 complex of CMN: PEG; two and 3D illustrate the structures of enol form of CMN on the monomer displaying hydrophobic interaction of phenolic OH group. (Green tubes denote the monomer piece of CMN, and Grey tubes denote PEG; dotted lines indicating the most significant bonds, the distances are revealed in Å units).

**Figure 2 pharmaceuticals-14-00939-f002:**
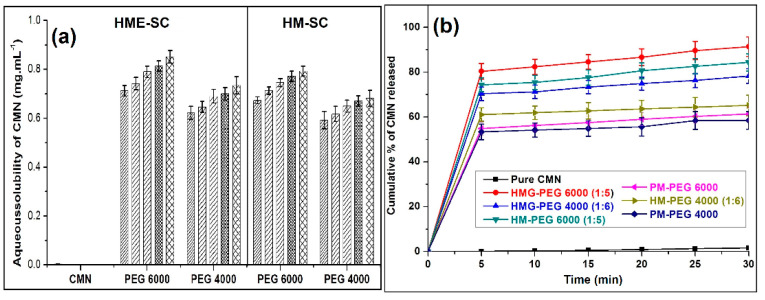
(**a**) Bar chart represents the solubility of CMN in HME and HM technology, and (**b**) dissolution release of curcumin from PM and HME and HM-SC (mean ± SD, n = 3).

**Figure 3 pharmaceuticals-14-00939-f003:**
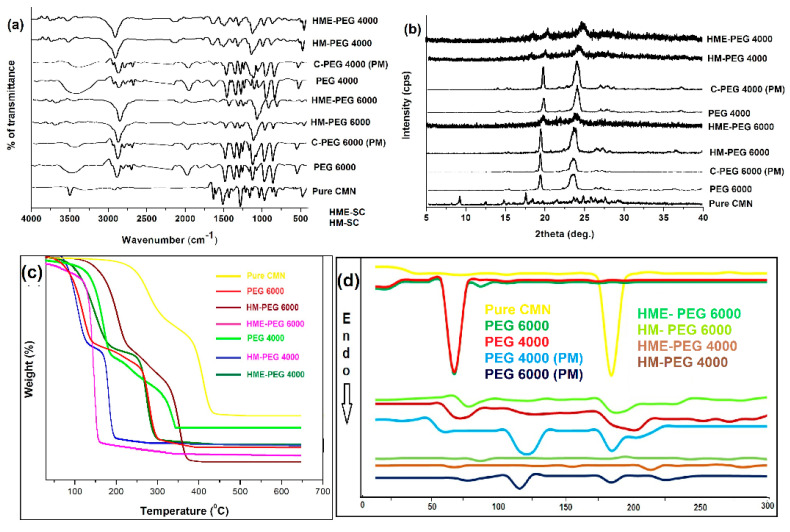
The FTIR spectrum (**a**), PXRD pattern (**b**), TGA curve (**c**), and (**d**) DSC thermogram of pure CMN, PMs, and SCs.

**Figure 4 pharmaceuticals-14-00939-f004:**
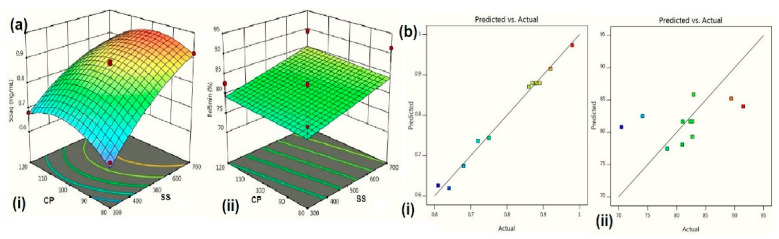
(**a**) 3D response surface from the CCD. (**b**) Plot of predicted versus actual response of (**ai**,**bi**) Solaq (mg.mL^−1^) and (**aii**,**bii**) Rel_5min_ (%) results from HME-PEG 6000 SC.

**Figure 5 pharmaceuticals-14-00939-f005:**
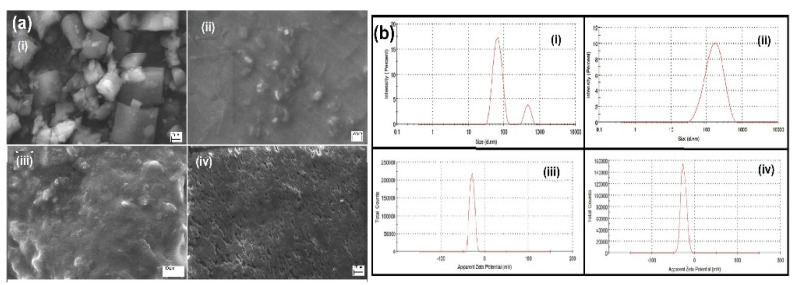
(**a**) SEM image of (**ai**) Pure CMN, (**aii**) PEG 6000, (**aiii**) HM-SC and, (**aiv**) HME-SC, and (**b**) DLS study; particle size of (**bi**) CMN-PM, (**bii**) CMN-SC, the zeta potential of (**biii**) HM-SC and (**biv**) HME-SC.

**Figure 6 pharmaceuticals-14-00939-f006:**
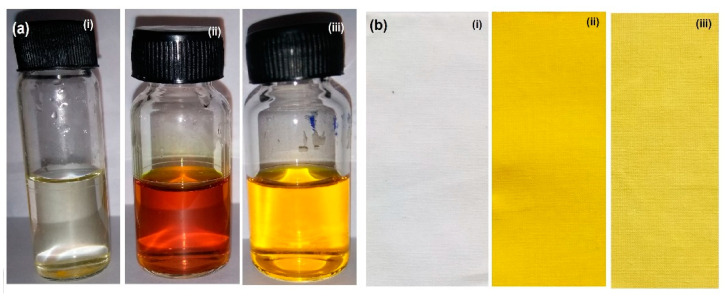
Photograph of (**a**) solution, (**b**) immersed cotton clothes of (**i**) pure CMN (**ii**) HME-SC, and (**iii**) HM-SC.

**Figure 7 pharmaceuticals-14-00939-f007:**
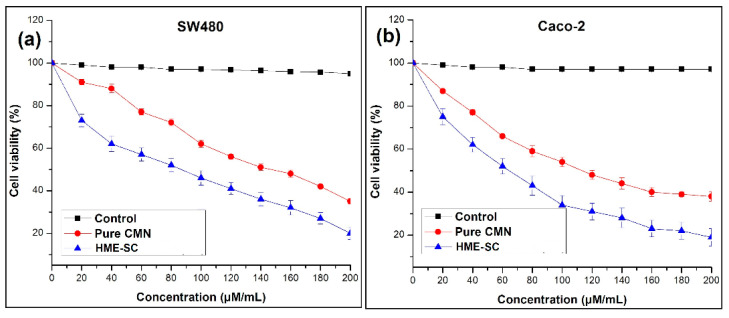
In vitro cytotoxic outcome of on (**a**) SW480 and (**b**) Caco-2 cell lines of control, pure CMN and HME-SC (mean ± SD, n = 3).

**Figure 8 pharmaceuticals-14-00939-f008:**
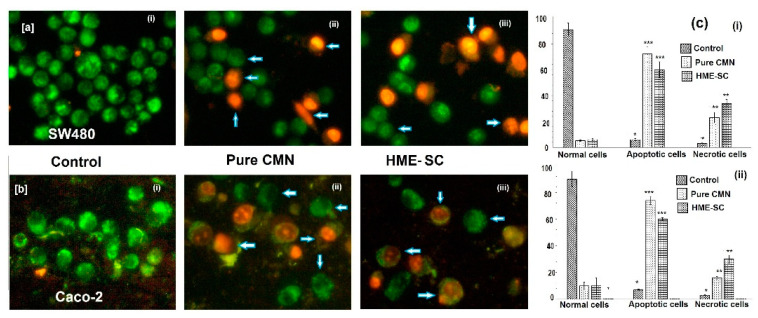
Apoptotic fluorescing image of AO/EB staining observed under fluorescent microscope with (**a**) SW480 and (**b**) Caco-2 cells; (**i**) control; (**ii**) pure CMN; and (**iii**) HME-SC and (**c**) % of live, apoptotic and necrotic cells after 24 h treatment (**i**) SW480 and (**ii**) Caco-2 cells. The significant differences associated to control are represented by *** *p* < 0.001 and ** *p* < 0.05, both are evaluated by Student’s t-test.

**Figure 9 pharmaceuticals-14-00939-f009:**
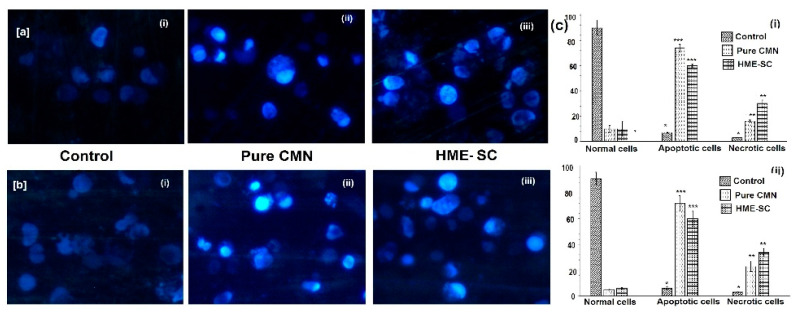
Apoptotic fluorescing image of Hoechst 33258 staining observed under fluorescent microscope with (**a**,**ci**) SW480 and (**b**,**cii**) Caco-2 cells; (**i**) control; (**ii**) pure CMN; and (**iii**) HME-SC and (**c**) % of live, apoptotic and necrotic cells after 24 h treatment (**i**) SW480 and (**ii**) Caco-2 cells. The significant differences associated to control are represented by *** *p* < 0.001 and ** *p* < 0.05, both are evaluated by Student’s *t*-test.

**Figure 10 pharmaceuticals-14-00939-f010:**
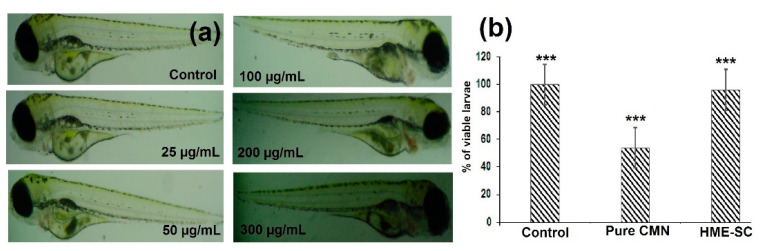
The in vivo toxicity studies of (**a**) photograph of the zebrafish larvae treated with various concentrations of HME-SC. (**b**) Graph representing the viable larvae (%) treated with pure CMN and HME-SC for 96 hpf. Significant difference associated to control are specified by *** *p* < 0.05 and were calculated with Student’s *t*-test.

**Table 1 pharmaceuticals-14-00939-t001:** Thermodynamic parameters of CMN with different carriers at 25 and 37 °C (mean ± SD, n = 3).

Carrier	T (°C)	Intercept (Mm)	Ka (M^−1^)	ΔG (kJ.mol^−1^)	ΔH (kJ.mol^−1^)	ΔS (kJ.molK^−1^)
**PEG 4000**	25	3.74 × 10^−4^	141.38 ± 6.842	−12.27 ±0.221	14.2 ± 0.762	0.08882 ± 0.002
	37	6.27 × 10^−4^	176.38 ± 7.874	−13.33 ± 0.234		
**PEG 6000**	25	4.61 × 10^−4^	309.79 ± 14.814	−14.22 ± 0.087	12.7 ± 0.567	0.09093 ± 0.001
	37	6.78 × 10^−4^	377.89 ± 15.846	−15.30 ± 0.067		

**Table 2 pharmaceuticals-14-00939-t002:** Formulation of SC by hot-melt extraction using central composite design.

Code	Coded Value		Sol_aq_		Rel_5min_	
			Y_1_		Y_2_	
	X_1_	X_2_	Actual	Predicted	Actual	Predicted
**P1**	300 (−1.00)	80 (-1.00)	0.642	0.635	80.96	79.10
**P2**	700 (1.00)	80 (-1.00)	0.924	0.915	91.47	88.02
**P3**	300 (−1.00)	120 (1.00)	0.681	0.675	82.71	81.30
**P4**	700 (1.00)	120 (1.00)	0.864	0.87	89.41	88.23
**P5**	217.157 (−1.41)	100 (0.00)	0.614	0.626	78.35	71.35
**P6**	782.843 (1.41)	100 (1.00)	0.985	0.974	82.93	83.85
**P7**	500 (0.00)	71.7157 (−1.41)	0.722	0.732	76.49	78.48
**P8**	500 (0.00)	128.284 (+1.41)	0.758	0.744	74.11	78.02
**P9**	500 (0.00)	100 (0.00)	0.881	0.880	82.62	81.66
**P10**	500 (0.00)	100 (0.00)	0.89	0.880	81.05	81.66
**P11**	500 (0.00)	100 (0.00)	0.882	0.880	82.48	81.66
**P12**	500 (0.00)	100 (0.00)	0.878	0.880	82.7	81.66
**P13**	500 (0.00)	100 (0.00)	0.889	0.880	82.36	81.66

## Data Availability

Data sharing not applicable.

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
