# Peer review of "Preparation of Soluble Complex of Curcumin for the Potential Antagonistic Effects on Human Colorectal Adenocarcinoma Cells"

_pharmaceuticals, 2021, doi:10.3390/ph14090939_

Round 1

Reviewer 1 Report

The authors present a study on the effect of a soluble curcumin complex prepared by hot melt and hot-melt extrusion method followed with phase solubility, in silico, aqueous solubility, drug release, and physicochemical study plus in vitro and in vivo cytotoxicity. The study brings novel and interesting results, nevertheless, I have following concerns which must be reflected.

Introduction: there is none about curcumin, it is just briefly mentioned in the brackets, the authors must redo the introduction so that they justify, why they have chosen curcumin, what biological effects does it have, mentioning both pros and cons - see e.g. the following reference in which it is discussed + it should be cited:
Sarco/Endoplasmic Reticulum Calcium ATPase Inhibitors: Beyond Anticancer Perspective. J Med Chem. 2020 Mar 12;63(5):1937-1963. doi: 10.1021/acs.jmedchem.9b01509. Epub 2020 Feb 17. PMID: 32030976.

Materials and methods:
PEG - define abbreviation
Why the cell lines were passaged in DMEM, when for SW-480 cells, Leibovitz's L-15 Medium is highly recommended and CaCo-2 cells should be maintained in Eagle's Minimum Essential Medium plus 20 % FBS not only 10 % FBS. By changing the cultivating conditions so much, you totally change the characteristics of the cells, this is simply wrong.
Moreover, why antibiotics were used during passaging? It can significantly affect the results as well as create some synergistic effects, etc. And, even worse, 20 mL od 1 % ATBs? 20 mL, really?
Further, at what passage were the cells used for the individual experiments? What was the procedure of passaging? It is fully missing in the methodology.
What exact type/grade of FBS was used? What chemicals, not stated here, were used for passaging, what plastics were used, etc.? It is so incomplete.
Please use space between the values and units - in the whole text!
Molecular modeling: state the PDB numbers as well as for other databases, so that is clear, what have you exactly used.
Preparation of PM - describe in detail, just citation is not enough, mostly it is not complete in the previous articles and it is not possible to reproduce.
Please do not use abbreviations in the titles of the chapters. It is totally unclear in some cases, what it means.
rpm - why is written at some places in caps lock and at others in small letters
Solid-state characteristics - again, describe how and what you have done. It is not enough to just say that it is explained in the previous article.
MTT assay: what is 5 x 103 cells.mL-1, it should be in superscript I guess
moreover, how could be the cell density maintained at this amount of cells? it is wrongly written, I guess that the cells were seeded at this density, not maintained
further, please do not write 200 µL.well-1, it is nonsense, such unit have never existed
what percentage of DMSO was used as a control?
I do not understand the assay description, so, you just seeded the cells and then, measured the MTT without any cell treatment (which is not stated/written in detail in the text)? it does not make sense
20 mL of MTT per a well of 96-well??? again, nonsense
how many replicates - both technical and biological - were done for the MTT assay? it is nowhere stated as well as the statistical description is missing
what software was used for the IC50 determination?
Microscopy - what NA of the objective? what camera? what software? what intensity and what exposure time, etc., etc. The methodology details are again fully missing. "as described in...ref. " is not enough, write it in detail, so others can reproduce it
again, the number of sample replicates is missing
Danio rerio - must be in italics

Results: MTT assay - why only 24 h was evaluated, when normally 72 h is done? these data should be added. The figure quality must be improved, it is almost unreadable. Plus please, significantly improve the figure legend.

Discussion: discussion is fully missing even though that on curcumin, there are hundreds and hundreds of articles. discussion must be added and all the results presented must be sufficiently discussed, this is and alfa and omega of each article and cannot be omitted

in vivo and in vitro are Latin terms, therefore, they must be written in italics

Conclusion: very general, not saying much, it should be rewritten so that it is clear what was exactly the contribution in detail, not just some general and even not very well-phrased statemets

Author Response

Dear Referee thank you for your positive comments to improve our article. We tried to reply to all the suggestions you raised and hopefully it will be satisfactory.

Reviewer 2 Report

This manuscript described the investigation of the effect of hot melt (HM) and hot-melt extrusion (HME) technology on the efficacy and toxicity of curcumin (CMN). The authors demonstrated that HME-SD produced significant improvement towards solubility, dissolution with ideal stability. It was also found out that HME-SD exhibited enhanced cytotoxicity. A few concerns for the authors to consider:

  1. There is no drug-to-carrier ratio clearly indicated through out the manuscript. The authors did say the physical mixture (PM) of varying composition 1:3 to 1:7, and the CMN (10% w/w) and PEG6000/PEG4000 (30-70% w/w). But what is the optimal drug-to-carrier ratio for this study?
  2. Hot melt extrusion technology is a thermal processing technique, which increases the risk of thermal degradation of drugs. While curcumin is a chemically unstable phenol, more data are needed to show that this compound is stable during the HME process, especially when we see significant changes in the FTIR spectra between the PM and HME/HM complexes.
  3. In session 3.2 In silico interactions, the perfect molecular modeling for the 1:1 complex of CMN:PEG may not present what is actually happening. It looks like far more than 1 molecules of PEG should be interacting with CMN.
  4. For all the in vitro and in vivo assays using pure CMN and HME-PEG6000, how did the authors calculate the concentration of HME-PEG6000 need to be explained. Maybe drug release experiment under the assay conditions?
  5. In the western blot experiments, it is interesting that the HME-SD treatment introduced extra cleaved caspase-3, caspase-9 and PARP compared with pure CMN in both cell lines. Any explanation for this?
  6. Wording in this manuscript need to be checked carefully. A few typos including page 8, line 338 “Figure 1D” should be “Figure 2B”; page 10, figure 3D, there are two “C-PEG 6000 (PM)”.

Author Response

(The authors gave the same response as above.)

Reviewer 3 Report

The manuscript submitted by Jamal et al., written well for the improved therapeutic activity of curcumin against tumor cells. However, it needs major revisions before acceptable for publication.

The title of the manuscript not exactly reflect the rationale of the work in the abstract. What is the need of the investigation must be added in the abstract?

The language of the manuscript very poor. It should be edited by native English speaker or English edit software for typo, grammar and syntactic errors.

The manuscript results discussed in empirical. The authors must be interpreted and discuss the results with previously published works of the curcumin.

In abstract,

  • line#20, change the hot melt to melt casting or melting method. Hot melt is not proper terminology for preparation of solid dispersion or soluble dispersions.
  • Change the abbreviation of SD to SC will be more appropriate.
  • Line # 23, CDD – delete from this line and make sure it’s CCD.
  • Line #23 - significant improvement, what is the level of significance and against what?
  • Write the units of solubility and reduce the value to one decimal.
  • Line#27, specify how to confirm the amorphous nature of the drug in the complex?
  • Write the HME-SD of the CMN
  • Write the IC50 values of control formulation in both cell lines.
  • Line#29 - Apoptosis study exhibits that cell death mainly by apoptosis with a small percentage of necrosis. This statement unclear. Rewrite
  • What is the effect of melt method complex formulation on cytotoxicity?

Main text

Line #37 and 38, both are same. Delete any one of the statements.

Cite the reference for line #39-41. Also include the updated statics on the effected population worldwide.

Define CMN and BCS in line#49 and use abbreviations later uniformly throughout the manuscript.

The following manuscript reported for the enhanced solubility and tumor activity of the CMN. What is the rationale for the selecting the CMN soluble complex? They are also synonym of solid dispersions. Why they called soluble complex?

https://www.ingentaconnect.com/content/ben/acamc/2020/00000020/00000016/art00003.

https://www.sciencedirect.com/science/article/abs/pii/S0168365919306170.

https://link.springer.com/article/10.1208/s12249-019-1349-4.

https://www.tandfonline.com/doi/abs/10.3109/10837450.2013.846374.

https://www.sciencedirect.com/science/article/abs/pii/S0378517303006318.

https://pubs.acs.org/doi/abs/10.1021/acs.molpharmaceut.7b00319.

https://www.sciencedirect.com/science/article/abs/pii/S0278691518309190.

https://www.sciencedirect.com/science/article/abs/pii/S0378517311011811.

https://link.springer.com/article/10.1208/s12249-011-9732-9.

https://link.springer.com/chapter/10.1007/978-3-319-11776-8_72.

The HME formulation of CMN also reported for sustained release and immediate release purpose as well.

https://www.sciencedirect.com/science/article/abs/pii/S0022354919308032.

https://www.sciencedirect.com/science/article/abs/pii/S0022354919308056.

https://www.sciencedirect.com/science/article/abs/pii/S030881461400140X.

CMN nanoformulations for colon cancer activity also reported

https://www.frontiersin.org/articles/10.3389/fphar.2019.00152/full.

https://www.ingentaconnect.com/contentone/asp/jbn/2009/00000005/00000005/art00001.

https://www.tandfonline.com/doi/abs/10.1080/09205063.2018.1541500.

Nanoparticles and other colloidal carrier systems had better targeting efficiency than HME based formulations. The recrystallization of the carrier also problem for the preparation of SD through HME and melt methods. What is the advantage of the HME formulations over nanoformulations. Justify?

Line#51, weak pharmacokinetics of this molecule, that is, the solubility in an aqueous solution is very low – what is meant by ‘weak’ pharmacokinetics. The authors already mentioned CMN BCS class-IV drug. delete the later part from the sentence.

Line #57, crystalline form to amorphous form – But the CMN solubility not discussed respect to crystalline nature. Rewrite the statement. Delete line#52-62. This is not related to the work. Very regular in nature.

Why the authors used solid dispersion in the main text. Soluble complex is the main concept of the work and also used SD as abbreviation in the abstract.

Line#68 - operational methods – change to delivery approach

Focus on how CMN used as therapeutic option in colon cancer treatment.

What is the rationale for selection of two methods for SD preparation not clear?

Line#117 - Moreover, the most effective formulation that has an anticancer potential compared to the pure CMN. Delete this statement.

Overall, rewrite the introduction with rationale of the work.

How the melt method SD showed nm size range? The simple melting of the drug didn’t change the particle size in nm range. Write the mechanism for reducing the particle size through melt method. Further, this data not correlated with the SEM images.

What is the reason for measuring the particle size of the SD. Explain?

Author Response

(The authors gave the same response as above.)

Reviewer 4 Report

1D. It appears that the rate of dissolution of pure CMN was found to be 1.62 % toward the 338 end of 30 min

Observation .   Here , and in the entire paragraph and in the entire paper, when you write about   “ rate of dissolution”, it is necessary to specify if the term refers to first, burst release phase, or to second, slow-release phase.

The rate of dissolution on SD with their carriers exhibits a high 341 burst release (48.73– 83.28 %) in the initial 5– 6 min

 Obs.  Delete “the rate of”

General observation. The correct term is “rate of release”. The most utilized term, short-cut term is  “rate of dissolution”.  It is more appropriate to use the same  term in the entire paper.

In fig 2b time is minutes, not hours, as is written on the axis

From the pattern of release due to the metastable supersaturation of the CMN in the 352 wet PEGs concentration in the moderate dissolution

Obs, Unclear statement. Much more, entire paper seems to me fuzzy .

Release into water is not biorelevant. It is not clear what the consequences of the formulation are on the pharmacokinetics of pf CMN. This issue is worth discussing.

Author Response

(The authors gave the same response as above.)

Round 2

Reviewer 1 Report

The authors have corrected and explained what was needed, the article can be now accepted for publication.

Author Response

Dear Reviewer thank you for your satisfaction from our revision. We really appreciate your time, you spared for our article improvement.

Reviewer 2 Report

The revised manuscript made a lot of modifications and I have no further questions for the authors.

Author Response

Dear Reviewer thank you for recommending statement as The revised manuscript made a lot of modifications and I have no further questions for the authors.

we have checked the MS thoroughly for English language check (from a native professor) and minor spell checks. Hopefully now the article will be with optimum corrected language and spells.

The changes can be traced as red in track changes in main MS.

Reviewer 3 Report

Manuscript modified as per the suggested edits and satisfactory.

Author Response

Thank you for your response as (Manuscript modified as per the suggested edits and satisfactory) on our revision.

Regards

Reviewer 4 Report

The authors responded to my comments. I do not know the observations made by the other reviewers but I saw that the paper has many additions. I think the paper can be published in this form.

Author Response

Dear Reviewer really we are boosted when we saw your comments as (The authors responded to my comments. I saw that the paper has many additions. I think the paper can be published in this form.) after we submitted the revised MS in light of your suggestions.